# Acute Inflammation in Tissue Healing

**DOI:** 10.3390/ijms24010641

**Published:** 2022-12-30

**Authors:** Amro M. Soliman, Daniel R. Barreda

**Affiliations:** 1Department of Biological Sciences, University of Alberta, Edmonton, AB T6G 2R3, Canada; 2Department of Agricultural, Food and Nutritional Science, University of Alberta, Edmonton, AB T6G 2R3, Canada

**Keywords:** inflammation, tissue repair, neutrophils, macrophages, chronic injuries, inflammatory mediators

## Abstract

There are well-established links between acute inflammation and successful tissue repair across evolution. Innate immune reactions contribute significantly to pathogen clearance and activation of subsequent reparative events. A network of molecular and cellular regulators supports antimicrobial and tissue repair functions throughout the healing process. A delicate balance must be achieved between protection and the potential for collateral tissue damage associated with overt inflammation. In this review, we summarize the contributions of key cellular and molecular components to the acute inflammatory process and the effective and timely transition toward activation of tissue repair mechanisms. We further discuss how the disruption of inflammatory responses ultimately results in chronic non-healing injuries.

## 1. Introduction

Throughout millions of years of evolution, multicellular organisms have adapted complex systems for recognizing and repairing injured tissues [1]. Reparative responses share common pathways and characteristics, where the inflammatory response is considered a cornerstone for effective tissue repair [2]. Upon injury, inflammatory cascades are initiated to clear pathogens and regulate subsequent healing events. Still, tight regulation of acute inflammation is required to avoid its excessive perturbations, which ultimately results in defective and delayed healing. We previously highlighted some of the molecular and cellular mechanisms regulating the induction and resolution of inflammation during cutaneous infection [3]. These mechanisms further influence other parts of the healing process and potentially determine the outcome.

Tissue repair is a complex biological reaction to an injury involving interactions between various immune and connective tissue cells. Together, these cells and several humoral factors accomplish sequential phases, comprising hemostasis, inflammation and proliferation, to restore damaged tissues [4]. Subsequent to tissue damage, there is a constriction of injured blood vessels followed by platelet activation and clot formation to stop bleeding [5]. Fibrin threads act as a scaffold for infiltrating leukocytes. Neutrophils are among the early cells relocating to the injury site, representing the first line of defense against infection [6]. Monocytes follow neutrophils, where they mature to macrophages [7]. The cellular migration is triggered mainly via activation of an acute inflammatory program that involves expression of several cytokines and chemokines [8]. Following the eradication of infection and tissue debris, anti-inflammatory mediators and growth factors are released to suppress inflammation and initiate the proliferative phase [9]. Several tissue repair events, including angiogenesis, granulation tissue formation and re-epithelialization, are initiated.

The newly formed granulation tissue and epidermis require an adequate supply of nutrients and oxygen to maintain homeostasis and to promote further healing. This is achieved by developing new blood vessels at the wound area. The process is known as neovascularization or angiogenesis, accomplished via activating local endothelial cells (ECs) lining the inner surface of neighboring blood vessels [10]. In response to hypoxia-responsive growth factors (e.g., VEGF), ECs migrate, proliferate and form new cell-to-cell junctions to develop new capillaries branching out from existing blood vessels. Angiogenesis is regulated by angiogenic molecules other than VEGF, such as platelet-derived growth factor (PDGF), fibroblast growth factor (FGF) and angiopoietins [11]. ECs show heterogeneity in response to these molecules by functioning as either lead tip or trailing stalk cells. Lead tip cells migrate towards angiogenic factors in response to positive and negative regulators. On the other hand, stalk cells preserve the structure of existing blood vessels [12].

During the proliferation phase, new connective tissue is formed concurrently with neovascularization. Fibroblast is the key cell accountable for constructing granulation tissue to fill in the wound gap. In response to various signaling molecules that are released from tissue-resident macrophages, platelets, keratinocytes and ECs (e.g., transforming growth factor (TGF)-β, EGF and PDGF) [13,14,15], fibroblasts proliferate, migrate and become pro-fibrotic, depositing ECM proteins [16]. Prior to laying ECM proteins by fibroblasts, they obliterate provisional matrix by secreting MMPs, to be substituted by granulation tissue rich in collagen, fibronectin, glycoproteins and proteoglycans [17]. Fibroblasts in wound repair demonstrate functional diversity, stemming from their contribution to many other healing events [18]. Fibroblasts exhibit heterogeneity depending on their activation status and throughout different developmental stages [19], resulting in significant phenotypic differences. These phenotypes mediate varying functions in wound repair, including re-epithelialization [20], secretion of growth factors, immunomodulation, ECM synthesis and organization [21].Keratinocytes differentiate and migrate from wound edges toward the center to cover wound surface. This is achieved by weakening of cell–cell connections and adhesion to basement membrane [22]. Cellular migration continues until keratinocytes from opposing edges come in contact, forming the basal layer that develops new adhesions with the underlying matrix [23]. Suprabasal keratinocytes proliferate to provide multiple overlying layers of keratinocytes [24]. Re-epithelialization is regulated via several cytokines and growth factors as well as crosstalk between keratinocytes and inflammatory cells (PMN and macrophages), fibroblasts and ECs [25]. For instance, keratinocytes were found to activate fibroblasts to release growth factors, thus promoting keratinocyte proliferation [15]. Indeed, regulation of proliferation phase of tissue repair is orchestrated by various cytokines and growth factors released by inflammatory cells such as macrophages [26]. It is further dependent on efficient clearance of pathogens and proper resolution of inflammation [27]. This is clearly observed in disrupted tissue repair associated with conditions of impaired immune responses and prolonged inflammation that are instigated by, for example, diabetes, aging and malnutrition [28,29]. This review discusses the significant contribution of acute inflammatory reaction’s various components and effectors to tissue repair. Recognizing these regulatory factors is critical for understanding physiology of the repair process and pathophysiology of chronic non-healing injuries commonly associated with their disruption.

## 2. Induction of Inflammation Phase of Tissue Repair: Innate Immune Responses

Following an injury, danger/damage-associated molecular patterns (DAMPs) [30], generated by necrotic cells, as well as pathogen-associated molecular patterns (PAMPs) [31], including conserved motifs of invading pathogens, are recognized by innate receptors on tissue-resident cells to trigger an acute inflammatory reaction [17] (Figure 1). As a result, various inflammatory mediators are released to promote leukocyte recruitment and regulate immune responses at the injury site [32].

### 2.1. Immune System Perception of Injury: The Role of DAMPs and PAMPs

DAMPs are either passively or actively released by injured host cells [33,34]. They include patterns that are usually isolated inside cells with limited extracellular exposure. DNA (genomic or mitochondrial), ATP and other peptides are released into the extracellular space following cell death or lysis [35,36,37,38]. These patterns provide self-injury detection strategies for the host to activate inflammatory responses in the case of sterile injuries or wounds with restricted PAMPs. Mechanistically, DAMPs function by directly binding to pattern recognition receptors (PPRs) on resident cells or indirectly by modifying extracellular matrix (ECM) molecules such as heparan sulfate, fibrillary protein and collagen to possess proinflammatory-stimulating properties [39,40,41]. In addition to DAMPs, hydrogen peroxide (H_2_O_2_), chemokines and other lipid mediators are released by damaged cells to deliver signals promoting innate immune functions such as leukocyte recruitment. For example, H_2_O_2_ provokes neutrophil migration toward the injury site for rapid pathogen clearance [42]. On the other hand, PAMPs, also known as microbe-associated molecular patterns (MAMPs), represent parts of different invading microbes (e.g., viruses, bacteria, parasites and fungi). They embrace LPS, microbial lipoproteins, β-glucan and double-stranded RNA [43,44,45,46,47]. Similar to DAMPs, PAMPs can act as ligands for several PPRs to launch acute inflammatory programs.

### 2.2. Activation of PPRs and Downstream Inflammatory Pathways

The injury-recognition system markedly relies on innate PPRs located on tissue-resident cells. PRRs include several types, such as Toll-like receptors (TLRs), nucleotide-binding oligomerization domain-like receptors (NLRs), retinoic acid-inducible gene I like receptors (RLRs) and C-type lectin receptors (CLRs) [48]. TLRs were found to play a vital role in initiating the inflammatory phase of tissue repair [49] via specifically binding a variety of ligands. For instance, TLR1, TLR2 and TLR5 can detect bacterial peptidoglycans and flagellin, while TLR3, TLR7 and TLR8 recognize single and double-stranded RNA of viruses [50]. Activated TLRs trigger nuclear factor-kB (NF-κB) and mitogen-activated protein kinase (MAPK) pathways via adaptor proteins such as TRAM, TRIF, MyD88 and TIRAP/Mal, thus upregulating the expression of proinflammatory cytokines, such as IL-1β and TNF-α in addition to other chemokines, lipid mediators and adhesion molecules [51,52]. Interestingly, IL-1β and TNF-α provide a positive feedback loop by directly activating NF-κB-facilitated gene expression, thereby intensifying inflammatory responses [53]. Other transcription-independent pathways are activated early at the injury site via, for example, Ca^2+^ influx and reactive oxygen species (ROS) to compensate for the delay in the induction of transcription machinery [26]. Intracellular Ca^2+^ levels increased substantially within a few minutes at wound edges and later at the center after an injury [54]. All these pathways contribute to initiating an acute inflammatory state.

### 2.3. Inflammatory Cytokines and Mediators

Cytokines are small proteins (~10 kDa) characterized by having the amino acid, cysteine in their structure. The two most important cytokines involved in tissue repair are (1) CC cytokines which have two adjacent cysteines and (2) CXC cytokines which contain two cysteines separated by another amino acid. Cytokine production is a complex biological process regulated by several activators and inhibitors depending on the stage of inflammation and other environmental factors [55]. Cytokines are essential for the induction, propagation and resolution of the inflammatory phase of tissue repair [56]. Moreover, they promote cellular recruitment and regulate their development, proliferation and functions during the healing process, as shown in Table 1. Notably, most tissue-resident cells, including parenchymal cells, fibroblasts, ECs, and immune cells, can produce various cytokines in response to DAMPs and PAMPs. Furthermore, recruited leukocytes accentuate the release of these cytokines under a proinflammatory condition [57]. The pleiotropic properties of inflammatory cytokines allow them to exert a wide range of functions. For instance, despite its critical role in the induction of acute inflammation and leukocyte chemotaxis, IL-6 was also found to promote resolution [58,59].

Other crucial mediators of inflammation were shown to influence the healing process. For instance, upon injury, mast cells are triggered to release histamine, which in addition to increasing vascular permeability, it stimulates proliferation of keratinocytes [60] and fibroblasts to activate collagen synthesis [61]. Moreover, histamine is essential for platelet and integrin activation [62]. Prostaglandin (PG)E2, another mediator released early during wound healing, significantly modulates inflammatory and reparative responses [63]. PGE2 augments VEGF expression, regulating angiogenesis [64], and activates M2 phenotype of macrophages, promoting tissue repair [65]. This was further supported by impaired healing and excessive scar formation induced by PGE2 inhibition [66]. On the other hand, thromboxane (TX)A2, released by activated platelets immediately upon injury, contributes to platelet aggregation necessary for hemostasis [67]. It further induces the synthesis of IL6 and PGE2 [68] and promotes angiogenesis by enhancing EC migration [69]. Being a critical chemoattractant, leukotriene B4 (LTB4) enhances recruitment of various immune cells to the injury site [70]. However, uncontrolled production and release of LTB4 delayed resolution of neutrophils in diabetic mice [71].

**Table 1 ijms-24-00641-t001:** Cytokines involved in tissue repair and their potential biological functions.

Cytokine	Receptor	Source	Functions
TNF-α	TNFR1 (p55) and TNFR2 (p75)	PMN, macrophages and mast cells	‣ Induction of acute inflammation and cellular recruitment [72,73] ‣ Increases synthesis of adhesion molecules to augment PMN recruitment [74] ‣ Promotes angiogenesis [75] ‣ Enhances keratinocyte proliferation and expression of adhesion molecules [76] ‣ Mast cell-released TNF-α activates DCs migration and maturation [77] ‣ Acts as a mitogen for fibroblasts [78] ‣ Stimulates expression of PAFR and EGF receptors, enhancing cellular migration and proliferation [79]
IL-1β	IL-1R1, IL-1R2 and IL-1RAcP (IL-1R3)	keratinocytes, PMN and macrophages	‣ Induction of acute inflammation and cellular recruitment [72,73] ‣ Augments producing other proinflammatory cytokines, including TNF-α and IL-6 [80] ‣ Increases fibroblast-secreted KGF and FGF-7 to promote keratinocyte migration and proliferation [56,81,82] ‣ Triggers skin stem cell proliferation and activates gamma delta (γδ) T cells [83] ‣ Stimulates myofibroblasts to produce proteases, degrading damaged ECM as well as facilitating fibroblast migration and proliferation in response to mitogens [84] ‣ Promotes in vitro fibroblast migration and proliferation [85] ‣ Increases the expression of VEGF (proangiogenic factors) and MMP-1 (enhances ECM degradation) [84] ‣ Decreases expression of ECM proteins (e.g., collagen) and myofibroblast differentiation [84]
CXCL8	CXCR1	Platelets, PMN and macrophages	‣ A potent chemoattractant of neutrophils [86,87] ‣ Upregulates integrins and PMN-endothelium interactions to facilitate diapedesis [88] ‣ Enhances antimicrobial mechanisms of PMN, such as ROS production and release of neutrophilic granules [89]
IL-6	gp130 and IL-6R	Myeloid cells, lymphocytes and fibroblasts	‣ Induction of acute inflammation and cellular recruitment [72,73,90,91] ‣ Induces Th2 and Th17 differentiation in CD4^+^ T-cells [92] ‣ IL-6-stimulated Th2 cell release IL-4 and IL13 to activate polarization of M2 macrophage [93] ‣ Promotes TGF-β expression [94] and re-epithelialization [95] ‣ Enhances fibroblast proliferation, activation and migration [96] ‣ Augments wound closure and granulation tissue formation in glucocorticoid-induced immunosuppressed mice [97] ‣ Stimulates fibroblast differentiation to myofibroblast [98] ‣ Activates fibroblasts, macrophages and keratinocytes to secrete VEGF, thus promoting angiogenesis [99]
IFN-γ	IFNGR1 and IFNGR2	Natural killer cells, plasmacytoid DCs and T cells	‣ Antiviral activities [74] ‣ Activates macrophage to produce proinflammatory cytokines and enhances phagocytosis [100] ‣ Regulates differentiation of CD4^+^ T cells into Th1 effectors [100] ‣ Inhibits angiogenesis and collagen deposition via downregulating TGF-β-mediated signaling pathways [101,102]
IL-10	IL-10R	Macrophages, DCs, PMN, mast cells and T cells	‣ Induction of anti-inflammatory responses and resolution of inflammation [103] ‣ Inhibits expression of proinflammatory cytokines, chemokines and adhesion molecules in macrophages and neutrophils [104] ‣ Suppresses NO and ROS production [104] ‣ Promotes migration and invasion of fibroblasts [103] ‣ Induces and maintains production of hyaluronic acid [105] ‣ Protects against excessive collagen deposition and reduces scar formation [106]
TGF-β	type II TGF-β receptor	Macrophages, keratinocytes, fibroblasts and platelets	‣ Antagonize PMN chemoattractants (e.g., IL-8) and suppresses migration of inflammatory cells to the injury site [107,108] ‣ Enhances the expression of ECM components such as collagen and fibronectin by fibroblasts [56] and inhibits various MMPs [109] ‣ Promotes angiogenic activities of endothelial progenitor cells [110] ‣ Augments keratinocyte migration and overall re-epithelialization [111] ‣ Stimulate transformation of fibroblasts to myofibroblasts through the acquisition of αSMA via SMAD-dependent and independent transcriptional activity [112,113]

TNF-α: tumor necrosis factor-alpha; TNFR: tumor necrosis factor receptor; PMN: polymorphonuclear leukocytes; IL: interleukin; KGF: keratinocyte growth factor; FGF: fibroblast growth factor; CXCL8: C-X-C motif chemokine ligand 8; TGF-β: transforming growth factor-beta; IFNγ: interferon-gamma; DCs: dendritic cells; NO: nitric oxide; ROS: reactive oxygen species; ECM: extracellular matrix; MMPs: matrix metalloproteinase; PAFR: platelet activating factor receptor; VEGF: vascular endothelial growth factor; αSMA: alpha smooth muscle actin.

### 2.4. Cellular Recruitment to Injury Site

A number of CC and CXC cytokines act as chemoattractant proteins, identified as chemokines, regulating the migration of several immune and non-immune cells that are critical for the repair process. More than fifty chemokines and eighteen chemokine receptors have been characterized in humans and mice [114]. Production and diffusion of chemokines have to be precisely controlled to tailor their availability, thus accurately directing migrating cells [115]. Unfortunately, mechanisms regulating chemokine synthesis and diffusibility are still largely uncharacterized. It is worth mentioning that particular chemokines are constitutively produced and released under normal conditions to maintain tissue homeostasis via regulating basal cell functions and trafficking [116,117]. Though, in response to an injury, chemokines are expressed at higher levels during an acute inflammatory reaction to execute necessary immune responses. CXC chemokines containing glutamate–leucine–arginine (ELR) motifs such as C-X-C motif chemokine ligand 8 (CXCL8), also known as IL-8, are more specialized in polymorphonuclear leukocyte (PMN) recruitment [118,119]. Meanwhile, other CC cytokines, including C-C motif ligand 1 (CCL1) act as monocyte [120] and lymphocyte [121] attractants. The expression of these chemokines must be firmly regulated during tissue repair to avoid dysregulation of inflammatory responses. Persistent uncontrolled expression of particular chemoattractants results in the development of a variety of pathological conditions [115,122,123,124,125,126]. Upon their release, chemokines bind to glycosaminoglycans on ECs of blood vessels to be presented to circulating immune cells [127]. Leukocytes bind to these chemokines via their corresponding G protein-coupled receptors (GPCRs), resulting in the extravasation of these cells and migration toward the injury site [128,129]. For instance, C-X-C motif chemokine receptor 1 (CXCR1) and CXCR2 on PMN bind to CXCL8, activating downstream signaling pathways and promoting neutrophil recruitment [130]. Intriguingly, as reviewed by Ridiandries et al. [95], several chemokines were shown to contribute not only to inflammatory cell recruitment but also to proliferation and remodeling phases of tissue repair. For instance, CXCL1 and CXCL7 were involved in angiogenesis [131], while CXCL12 promotes differentiation of stem cells into fibroblasts and ECs, enhancing granulation tissue formation [132,133].

Leukocyte extravasation is achieved through several steps involving adhesion molecules (e.g., selectins and integrins), chemokines and interactions with ECs. Selectins (E-, P- and L-selectins), type I transmembrane proteins, are principally responsible for the initial tethering and adhesive interactions between ECs and circulating leukocytes [134]. Therefore, these adhesion molecules, such as E-selectin, were found to be upregulated by proinflammatory cytokines [135]. Selectins bind to carbohydrate-based ligands such as P-selectin glycoprotein ligand-1 (PSGL-1), generally expressed on leukocyte microvilli, to secure these cells to ECs at the injury site, where they mediate most tethering and rolling [136]. Integrins, on the other hand, are expressed on leukocytes, where they get activated by proinflammatory cytokines to induce cellular adherence to counter-receptors, such as intercellular adhesion molecules (ICAMs), enhancing adhesion of circulating immune cells to endothelium [137,138]. During leukocyte recruitment, various integrins are activated during different steps of transendothelial migration, where they were also found to support cellular arrest and rolling [139]. The integrin-counter receptor contact stimulates signaling cascades in captured leukocytes to achieve an intermediate-affinity conformational change, resulting in the “slow rolling” of these cells [136,140]. Pathways involved in integrin activation of leukocyte rolling, including inside-out and outside-in signaling mechanisms, are reviewed by McEver et al. [140]. Macrophage-1 antigen (Mac-1), a part of integrin cell surface receptors, was reported to be a crucial mediator of monocyte rolling with the help of P-selectin on low-shear stress ECM substrates [141]. Integrins can also stimulate cytokine secretion directly in neutrophils and macrophages [142]. Leukocytes roll along the surface of the endothelium to sense glycosaminoglycan-bound chemokines. Activation of chemokine receptors on leukocytes results in confirmational changes and leukocyte adhesion cascade (diapedesis) [139,143,144], where inflammatory cells crawl through endothelial junctions or weak regions of the basement membrane [145]. In a typical repair process, there is usually an early expression of neutrophil chemoattractants, resulting in rapid recruitment of PMN to the injury site. This is followed by subsequent waves of infiltrating monocytes and lymphocytes triggered by other chemokines [146].

In addition to the conventional pathway of inflammation-induced upregulation of chemokines gene expression, another recently reported alternative mechanism for leukocyte recruitment could be attained by mast cells. These cells contain granules filled with proinflammatory cytokines, vascular permeability and vasodilation factors, as well as proteases. Immediately after an injury, these molecules are released to enhance migration of immune cells to the injury site [147]. Interestingly, mice with a deficiency in mast cell proteases showed remarkably impaired neutrophil recruitment [148]. These findings do not necessarily abolish the significant role of chemokine-mediated PMN migration; nevertheless, it provides an early pathway for neutrophils to exist rapidly at the injury site for early pathogen clearance and compensate for a possible delay in the activation of transcription machinery.

## 3. Role of Inflammatory Cells during Tissue Repair

### 3.1. Neutrophils

Among immune cells involved in the repair process, neutrophils are considered the “first responders” since they are swiftly recruited [149,150], constituting approximately 50% of all cells at the injury site within 24 h after injury [151]. In addition to the potent CXCL8, other cytokines, such as CXCL4 and CCL3/4 promote PMN migration [151]. Notably, neutrophils are not commonly detected in healthy skin; instead, they remain in the bone marrow and bloodstream [152], ready to be drafted, as discussed in the previous section. Recruited neutrophils can augment additional PMN infiltration by releasing several chemoattractant factors [150,153,154]. The primary function of neutrophils at the injury site is to combat invading pathogens via various antimicrobial responses, including phagocytosis, toxic granules, oxidative burst and neutrophil extracellular traps (chromatin filaments released extracellularly to immobilize and eliminate microbes, known as NETs) [149,155]. Still, a critical balance must be maintained between phagocytes’ protective functions and their possible contributions to prolonged and exacerbated inflammation [156]. This equilibrium ensures the eradication of infection while minimizing collateral tissue damage. Several studies suggested that the prolonged existence of neutrophils at the injury site was detrimental to proper tissue repair [6,157]. This was attributed to PMNs-derived proteases degrading ECM and being allied with a deleterious oxidative burst [158,159]. Recently, neutrophils were also found to induce genomic instability via ROS-independent pathway involving the release of microparticles containing proinflammatory microRNAs (miR-23a and miR-155) in patients with inflammatory bowel disease (IBD) [160]. These miRNAs promoted the accumulation of double-strand breaks (DSBs) by inhibiting homologous recombination (HR), resulting in impeding inflammation resolution and overall intestinal healing [160].

Neutrophils can engulf bacteria and tissue debris through phagocytosis. Although the process is similar to that of macrophages, PMN possess distinctive phagocytic receptors [161]. As a result, PMN could handle antigens differently, where they can be opsonized or not. Fc receptors such as CD16, CD32 and CD64 recognize pathogens and then activate downstream Src and Rho-GTPases phagocytosis pathway. The result is an extension of the cell membrane to surround the antigen, forming phagocytic cups that get sealed to form phagosomes [162]. Additionally, neutrophils are characterized by having distinct granules containing bactericidal agents. These granules either fuse with phagosomes to destroy the pathogen intracellularly [163] or undergo exocytosis to combat microbes extracellularly [152]. Antimicrobial agents of these granules include myeloperoxidase, lysozyme, matrix metalloproteases (MMPs), lactoferrin and proteases (e.g., elastase and capthepsin G) [6]. The utilization of proteases by neutrophils is not limited to their anti-bacterial activity. Proteases are likewise crucial for neutrophil extravasation via degrading ECM elements and basement membrane of ECs directly or indirectly by activating MMPs [164]. Despite their importance for PMN migration and bactericidal actions, an unrestrained increase in proteoses induces extensive tissue damage, ensuing impaired healing and chronic wounds. This is on top compounded by proteolytic enzymes-induced obliteration of growth factors, newly formed blood vessels and granulation tissue [165]. Among these proteoses, elastase is released in response to bacterial infection in either free or membrane-bound form, and it was found to induce ECM destruction and direct epithelial damage [166]. Soluble elastase causes damage to areas surrounding neutrophils at degranulation site, while membrane-bound elastase can travel distally, resisting inhibition by anti-proteinases [167]. Elastase further contributes to neutrophil disintegration and NETs release by translocating to the nucleus and degrading chromatin through splitting histones [155]. Elastase can upregulate chemokines (e.g., IL-8) and other proteinases (MMP-9), instigating a vicious cycle of neutrophile recruitment and inflammation-associated collateral tissue injury [168]. Moreover, elastase degrades T-cell receptors and blocks antigen presentation, thus impairing lymphocytic functions [169].

Recent experimental evidence suggested parallel immunomodulatory functions of neutrophils during tissue healing in addition to their bactericidal actions. This was observed in mice with myocardial infarction (a sterile injury model), where researchers have characterized N2 neutrophils to play a potential role in restoring injured tissue irrespective of their antimicrobial functions [170]. Mechanistically, neutrophils modulate macrophage phenotype from a proinflammatory to anti-inflammatory/reparative state following engulfing apoptotic PMN by these macrophages, a process is known as efferocytosis [171,172]. Modulated macrophages release proresolution cytokines (e.g., IL-10) and growth factors such as transforming growth factor-beta (TGF-β) to control inflammation and initiate healing of damaged tissue [171]. A genetically modified reduction in PMN recruitment (CXCR2^−/−^) in injured mice resulted in delayed re-epithelialization at wound sites [173]. Likewise, aging-induced delayed wound healing was postulated to be instigated by the downregulation of neutrophil numbers or functions in mice [174] and humans [175,176]. This was attributed to impaired neutrophil-tempted pathogen clearance and, therefore, late resolution of inflammation. Conflicting data showed an accelerated re-epithelialization during neutrophil depletion [177]. Even though differences were observed at the level of epidermis development, no significant changes were evidenced in the dermis in terms of collagen deposition [177]. Still, further research is encouraged to characterize other possible immunomodulatory functions of PMN during tissue repair.

### 3.2. Macrophages

Macrophages play a critical role in tissue repair stemming from influencing both the inflammatory and proliferative phases. Macrophages’ contributions to immunomodulation, resolution of inflammation and tissue healing have been well-studied [178]. Macrophage numbers increase gradually at the injury site and peak 48–72 h after injury [179]. Influenced by chemokines such as CCL1 and CXCL12 [180], monocytes migrate to the injury site from bone marrow and adjacent blood vessels. Additionally, recruited macrophages can amplify the relocation of additional monocytes via releasing monocyte chemoattractant protein (MCP)-1 [181].

Several macrophage phenotypes were characterized during tissue repair [182,183,184]. It is worth mentioning that these phenotypes are not distinctively represented by particular macrophage subsets or a subject of on/off switch but rather a dynamic continuation of macrophage polarization based on environmental stimuli and interplay with other cells [185,186]. For instance, during the early phases of tissue repair, a classically activated macrophage phenotype, also known as M1, was shown to induce proinflammatory and bactericidal activities via expressing IL-1β and TNF-α in addition to mediating phagocytosis [187]. Later during the repair process, macrophages transition to becoming alternatively activated (M2) macrophages to suppress inflammation and promote the healing of damaged tissues [182]. Interestingly, recent reports indicated that M2 phenotype activation has expanded to involve other phenotypes triggered by various stimuli, such as M2a, M2b, M2c and M2d [184]. For example, M2a is activated by IL-4 and IL-13, while exposure to IL-10 and glucocorticoids stimulates M2c phenotype [184]. These M2 phenotypes largely intercede in anti-inflammatory, proresolution and healing roles [188]. Notably, macrophage phenotypes are not limited to the previously mentioned categories. There are likely several other phenotypes that are continuously activated depending on the differentiation stage, type and duration of stimulus as well as overall biochemical milieu [186,189].

The expanding literature supports the crucial role of macrophages in normal tissue repair. For instance, depletion of macrophages in wounds of murine models was associated with delayed healing induced by impaired angiogenesis, collagen synthesis and growth factors expression [190,191,192], indicating a significant engagement of macrophages in various repair events. Table 2 summarizes the potential functions and contributions of macrophage phenotypes during tissue healing.

### 3.3. Dendritic (DCs) and Langerhans Cells (LCs)

Since the discovery of DCs by Ralph Steinman in 1973, their role in immune responses and tissue homeostasis has been widely examined [209,210,211]. DCs can generally be categorized into tissue-resident and circulating DCs, also known as plasmacytoid DCs [212]. Tissue-resident DCs reside in tissues for immune surveillance, while plasmacytoid DCs are commonly absent in healthy tissues and are recruited following an injury [213,214]. The primary mission of DCs is to function as antigen-presenting cells that deliver antigens to T cells (CD8^+^ T cytotoxic and CD4^+^ T helper cells), establishing a bridge between innate and adaptive immunity [215]. The role of DCs in tissue repair was investigated mainly in murine models of wound healing, where DCs were found to be crucial for normal reparative responses. For instance, depletion of the early infiltrating plasmacytoid DCs after injuries considerably impaired the expression of proinflammatory cytokines and delayed re-epithelialization during wound healing in mice [213]. Moreover, a significant reduction in wound closure rate and granulation tissue deposition was observed in transgenic mice with depleted DCs [216]. Mechanistically, researchers suggested DCs to promote fibroblast proliferation and collagen synthesis in burn wounds. Recent evidence indicated a cross-talk between DCs and epithelial cells to maintain tissue homeostasis and promote tissue repair in the cornea. During corneal wound healing, DCs were shown to migrate with epithelial sheets to cover the wound surface [217]. On the other hand, the depletion of DCs resulted in an interruption of wound closure. Moreover, it significantly reduced epithelial cells-expressed CXCL10, IL-1β and thymic stromal lymphopoietin [217].

In the skin, DCs are identified as Langerhans cells (LCs), constituting approximately 2% of the epidermal cells [218]. Following an injury, LCs, similar to DCs, conduct antigen presentation after phagocytosing pathogens [218]. LCs can extend their dendrites through cellular tight junctions or completely reposition to reach microbes [219], inclined by cytokines that are primarily secreted by adjacent keratinocytes (e.g., MCP-1) [220]. Upon engulfing antigens, LCs translocate from epidermis to dermis layer by downregulating e-cadherin expression and utilizing the MMPs-induced degradation of basement membrane and ECM [221,222]. Guided by chemokines such as CXCL12, LCs migrate to draining lymph nodes, activating cell-mediated adaptive immune responses [223]. LCs can also keep tissue homeostasis by enhancing the activation and proliferation of T regulatory cells (T_reg_) [224,225]. Moreover, a significant role of LCs in diabetic wound healing was reported, where high numbers of LCs in diabetic foot ulcers were associated with a better healing outcome [226].

### 3.4. Mast Cells

Mast cells are specialized secretory cells differentiating from their precursors that migrate from bone marrow to perivascular regions of various connective tissues [227]. Since their discovery, mast cells have been recognized for their essential role in allergic reactions and combating parasitic infestations. However, many investigators recently became interested in studying their contributions to tissue repair. The growing evidence indicates mast cells to be critical for all phases of the repair process; hemostasis, inflammation and proliferation, as shown in Table 3. This is achieved primarily via cross-talk between mast cells and several other cells involved in tissue healing, modulating and triggering various activities [228]. Mast cell numbers increase at the injury site to reach about fivefold their original numbers, which was suggested to be a chemokine-mediated migration rather than cellular proliferation [151]. Despite the lack of studies examining chemokines regulating the migration of mast cells, keratinocytes-secreted MCP-1 was proposed to facilitate mast cell recruitment [151,229]. Mast cells are activated subsequent to binding of their receptors to a variety of ligands [230]. Upon activation, they release three different categories of molecules [231]. These molecules include (1) mediators that are constitutively stored in granules: histamine, serotonin, tryptase and heparin; (2) mediators that are synthesized in response to stimuli: leukotriene B4 (LTB4), prostaglandin (PG)D2 and lipid mediators; (3) cytokines and growth factors: TNF-α, IL-1β, IL-5, IL-8, granulocyte-macrophage colony-stimulating factor (GM-CSF), IL-10, VEGF and TGF-β [232,233,234,235]. Even with the current evidence on the significant role of mast cells in tissue repair, these cells showed heterogeneity in their functions depending on the tissue they reside in [236]. In addition, their phenotype changes according to the surrounding microenvironment [237]. Therefore, diverse mast cell subtypes with distinctive functions are yet to be characterized.

### 3.5. T Cells

Particular subtypes of T cells were shown to play a crucial role in tissue healing. For instance, dendritic epidermal T cells (DETCs) enhance re-epithelialization and granulation tissue formation via releasing various cytokines and growth factors [251,252]; consequently, mice with depleted DETCs experienced delayed wound closure and impaired ECM deposition [252,253,254]. Mechanistically, DETCs are activated by ligands (e.g., Semaphorin 4D and SKINTs) that are secreted mainly by keratinocytes upon injury [254,255]. Interestingly, stimulated DETCs express cytokines that are involved primarily in promoting keratinocyte proliferation, such as insulin growth factor-1 (IGF-1) and keratinocyte growth factors (KGF) [252,256]. Yet, the role of DETCs in several other tissue repair events is still undetermined.

Another type of T cell that is engaged directly in tissue repair is T_reg_. Several subsets of T_reg_ were detected in various peripheral non-lymphoid organs [257]. The core functions of these cells are to negate detrimental inflammation and maintain tissue homeostasis [258,259]. T_reg_ was found to reside in human and murine healthy skin, indicating their potential role in immunosurveillance and reparative responses in cases of injuries [260,261,262]. T_reg_ predominantly mediates immune suppressive activities during tissue repair, where they contribute to suppressing inflammation and the transition toward the proliferative phase [263,264]. The immune-suppressive actions of T_reg_ were attributed to their inhibition of IFN-γ^+^ T effector cells and proinflammatory Ly6C^+^ monocytes [265]. Furthermore, T_reg_ directly regulates the polarization of anti-inflammatory/reparative M2 macrophages via expressing IL-13 and IL-4 [266] in addition to enhancing efferocytosis [267].

## 4. Suppression of Inflammation

Following eradication of infection and exclusion of cellular debris, a transition towards an anti-inflammatory program is essential for activating reparative pathways that restore the structure and function of damaged tissue. The process is achieved through a variety of suppressive signals prompting a reduction in proinflammatory mediators and infiltrating leukocytes in addition to upregulation of proresolution molecules, including IL-10 and TGF-β [268]. Several pathways were reported to regulate the resolution of inflammation, and they rely primarily on effectively eliminating microbes [269]. Defects in pathogen clearance necessitate a continuation of a proinflammatory reaction that ultimately results in delayed healing. The process further involves a cross-talk and interplay between immune and non-immune cells at the injury site.

Pathways regulating the control of inflammation can be categorized into cell- and cytokine-mediated mechanisms. The launch of resolution of acute inflammatory responses is likely to be timely mapped with the fading of PMN from the injury site [26]. Downregulation of PMN can be achieved through two main mechanisms: (1) apoptosis followed by efferocytosis and (2) reverse migration. We previously highlighted the role of neutrophils in resolving inflammation via their efferocytosis by macrophages [270]. The process is accomplished by binding cellular communication network factor 1 (CCN1), present on phosphatidylserine of apoptotic PMN, to integrins of macrophages [271]. Engulfing apoptotic neutrophils is critical to avoiding their secondary necrosis, which leads to a substantial release of detrimental proinflammatory cytokines and ROS [270], and subsequently transforming macrophages into an anti-inflammatory phenotype [135] (Figure 1). On the other hand, recent data indicates a retrograde migration of neutrophils back into circulation as a pathway of PMN resolution [150,272] (Figure 1). This was shown in various models of mice, zebrafish, and in vitro human neutrophils. Notably, prolonged inflammatory conditions have been commonly associated with extensive and prolonged neutrophil infiltration, resulting in chronic wounds [273]. Additionally, M2 macrophages control inflammation by secreting various anti-inflammatory mediators (Table 2). Other cell types at the injury site were reported to express proresolution cytokines such as IL-4, IL10, IL-13, IL-35 and TGF-β [274,275]. These cytokines suppress inflammation by inhibiting the synthesis of proinflammatory cytokine and chemokine [269,276]. Furthermore, they reduce cellular infiltration by repressing the expression of adhesion molecule and diminishing chemokine-mediated leukocyte recruitment [269,277].

## 5. Dysregulation of Inflammatory Responses and Its Outcome

Tight regulation of acute inflammation is critical for normal tissue repair. We previously emphasized the significance of cellular and cytokine effectors in the induction of acute inflammation. Dysregulation of these inflammatory responses eventually disrupts the healing process due to failure to transition to the proliferative phase (Figure 2). This involves either impairment or exaggeration of inflammatory mechanisms such as leukocyte recruitment and production of proinflammatory mediators. Therefore, a balance has to be maintained and resolution must be achieved in a timely manner to avoid extravagant inflammatory responses and their associated collateral tissue damage.

Non-healing injuries are commonly accompanied by persistent inflammation. Mechanistically, several factors were described to explain this phenomenon. For example, deregulated proteolytic activities (e.g., overproduction of proteoses) in uncontrolled inflammatory reactions can devastate protective repair mechanisms, including cleaving growth factors [278,279,280]. Activity and expression of various MMPs were substantially upregulated in chronic wounds [281]. Another factor is extensive and persistent neutrophil infiltration. Compromised resolution of PMN at the injury site is escorted by detrimental levels of ROS and proteases inducing damage to cell membranes, ECM and crucial tissue repair mediators such as TGF-β and PDGF [282,283]. Likewise, macrophages in chronic wounds are associated with reduced levels of tissue inhibitors of MMPs (TIMPs), thus augmenting ECM degradation and delaying healing [284]. Macrophages also tend to present a dysregulated expression of inflammatory mediators and growth factors in non-healing injuries [9]. This is further complicated by an imbalance in M1/M2 phenotype where alternatively activated macrophages are significantly diminished [285]. Conversely, Keratinocytes show impaired migration and proliferation abilities in chronic injuries [286,287,288]. Likewise, fibroblasts suffer the loss of their proliferative potentials due to being less responsive to growth factors [289].

Based on the previously mentioned observations, it is generally agreed that a proinflammatory cycle must be broken in order for non-healing injuries to heal properly. Therefore, to develop therapeutic measures that shield restored tissue from the detriment induced by persistent inflammatory microenvironments, it is crucial to unravel and fully characterize pro- and anti-inflammatory pathways in tissue repair [290].

## 6. Current Tissue Engineering Strategies Managing Chronic Injuries

During the past two decades, various therapeutic strategies employing tissue engineering technologies were introduced in the field of tissue repair as reviewed by Yu et al. [291]. These applications advanced progressively to include auto/allografts, engineered skin grafts, cell-based therapy, cytokine/growth factor delivery and modern multifunctional biomaterial-based dressings such as carbon nanomaterials, hydrogel, fibrous scaffold, sponge, acellular dermal matrix and foam. Herein, we focused on strategies modulating the inflammatory milieu and immune system behaviour through molecule analogs and signaling ligands released locally by engineered constructs. In chronic wounds, M1 (proinflammatory) phenotype fails to polarize to M2 (anti-inflammatory), leading to the persistent release of TNF-α and IL-1β, thus maintaining a state of chronic inflammation [202]. One of the currently utilized strategies is to activate polarization of these cells. For example, monocyte and macrophages, when exposed to sphingosine-1-phosphate in vitro, they preferentially retain anti-inflammatory phenotypes [292,293]. A polyvinyl alcohol sponge implant injected with sphingosine-1-phosphate and ciclopirox olamine (antifungal) demonstrated proangiogenic properties in diabetic rats [294]. Polymers such as poly lactic-co-glycolic acid with the capacity to control release of fingolimod enhanced recruitment of anti-inflammatory M2 macrophages and monocytes through stromal cell-derived factor-1 alpha-mediated chemotaxis [295]. Likewise, the properties of Keratin biomaterials and dextran isocyanatoethyl methacrylate ethylamine hydrogel were found to influence immune cell behaviour and responses, including M2 polarization [296,297]. These attributes were combined with promotion of overall wound healing and hair follicle formation.

Others considered using natural remedies that directly inhibit the chronic inflammatory cycle. For instance, a cellulose nanocrystal film releasing curcumin, a polyphenol with anti-inflammatory and antimicrobial effects, was found to enhance bacterial clearance and overall wound healing in diabetic rats [298]. Similarly, fabricated chitosan-sodium hyaluronate-resveratrol sponges sustaining resveratrol release into wounds promoted neutrophil resolution, granulation tissue formation, re-epithelialization and angiogenesis [299]. Directly inhibiting inflammatory signals locally could be a potentially effective strategy in managing chronic wounds. Nanoparticles associated with siRNA were reported to significantly reduce TNF-α and MCP-1 production by macrophages and fibroblasts, respectively [300]. These strategies avoid systemic anti-TNF-α therapeutic applications that risk the development of global immunosuppression.

Surface chemistry and topographical patterning, among other biomaterial modulations, can alter leukocyte microenvironments and their phenotypes [301]. The engineering of hydrogel with various peptide motifs creates immunomodulatory scaffolds that influence inflammatory responses at the injury site. For instance, fabricated hydrogel containing bioactive peptides (e.g., substance P) was reported to recruit mesenchymal stem cells (MSCs) that modulate inflammation intensity and promote T_reg_ generation [302]. Future research aims at characterizing modulatory effects of several materials and biochemical factors may open the door toward robust therapeutic applications for chronic injuries.

## 7. Concluding Remarks

Despite the critical role of acute inflammation in tissue healing, a delicate regulation of its complex interacting network of diverse leukocyte subsets along with various pro- and anti-inflammatory mediators has to be maintained for efficient restoration of tissue homeostasis devoid of extensive collateral tissue injury. Importantly, a shift between the inflammatory and proliferative phases is crucial to preclude the unnecessary persistence of inflammation at the injury site. PMN apoptosis and their efferocytosis by macrophages are among the key signals inducing this transition. Pathologic and chronic non-healing injuries are mainly caused by the dysregulation of critical cellular and cytokine effectors during an acute inflammatory reaction, which lessens the effectiveness of the healing process. Therefore, it is necessary to uncover more about the molecular and cell-mediated mechanisms that regulate the inflammatory phase of tissue repair. This will provide insights that may open the door to novel therapeutic applications and strategies achieved via fine-tuning or enhancing these inflammatory pathways.

## Figures and Tables

**Figure 1 ijms-24-00641-f001:**
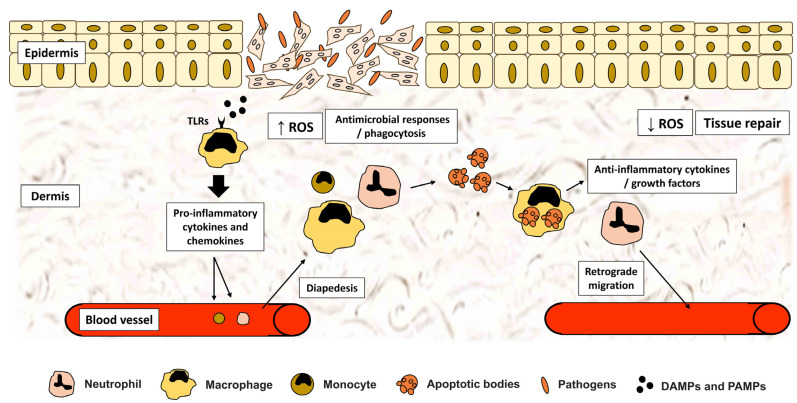
Induction and resolution of acute inflammation during tissue repair. Pathogen/Damage Associated Molecular Patterns (PAMPs/DAMPs) associated with an injury bind to Toll-like receptors (TLRs) expressed by tissue-resident cells, including macrophages. These cells release proinflammatory mediators and chemokines to activate an acute inflammatory program, recruiting leukocytes from nearby blood vessels. Neutrophils and monocytes gradually infiltrate the injury site to exert antimicrobial mechanisms, including increased reactive oxygen species (ROS) production. Following the eradication of pathogens, neutrophils undergo apoptosis to be engulfed by macrophages (efferocytosis). Activated macrophages endure polarization to release anti-inflammatory cytokines leading to resolution of inflammation and reduction in ROS levels. Moreover, a retrograde migration of neutrophils aids in decreasing infiltrating leukocytes. Additionally, macrophages release various growth factors to trigger tissue repair machinery.

**Figure 2 ijms-24-00641-f002:**
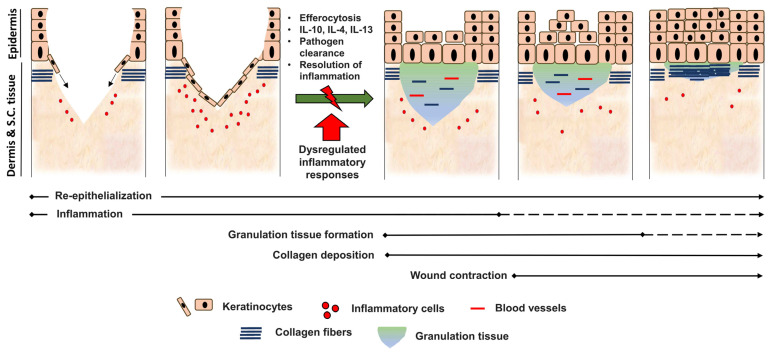
A classic wound healing model shows the importance of transitioning from the inflammatory to proliferative phases during the repair process. Normal wound healing typically progresses via activation of an innate immune program involving inflammatory cell recruitment. Anti-inflammatory responses are triggered through several mechanisms, including efferocytosis, to suppress inflammation and initiate repair events. These reparative phases comprise re-epithelialization, granulation tissue formation, angiogenesis and collagen deposition. Dysregulation of inflammatory responses in case of, for example, diabetes, aging and immunosuppressive diseases fail to transition to the proliferation phase, thereby inducing delayed wound healing or chronic injuries.

**Table 2 ijms-24-00641-t002:** Functions of macrophage phenotypes during tissue repair.

Phenotype	Receptors	Functions
M1 (classically activated or proinflammatory)	CD68 CD86 CD80	‣ Induces microbicidal activities (NO, ROS and phagocytosis) [187]. ‣ Releases proinflammatory cytokines (TNF-α, IL-1β and IL-6) [178,193] ‣ Enhances neutrophil recruitment by expressing chemokines [193,194] and synthesizing MMPs to degrade ECM [195] ‣ Clears apoptotic and necrotic PMN [196]
M2a (alternatively activated or wound healing)	CD163 CD206 CD209Ym1	‣ Activated by IL-4/IL-13 [197] ‣ Produces chemokines: CCL17, CCL18 and CCL22 as well as growth factors: IGF-1, fibronectin, TGF-β and PDGF [198,199] ‣ Promotes ECM formation and angiogenesis [200]
M2b (regulatory or type 2)	CD86	‣ Activated in vitro by phagocytosing apoptotic neutrophils [201] ‣ Inhibits inflammation by releasing IL-10 [201] ‣ Expresses IL-6, CCL1 and high levels of iNOS [198,202]
M2c (pro-resolving or deactivated)	CD86 CD163 CD206	‣ Stimulated by IL-10 via STAT3 pathway ‣ Releases IL-10 and TGF-β to exhibit anti-inflammatory responses [9,203] ‣ Expresses Mer receptor tyrosine kinase (MerTK), essential for efferocytosis [204]
M2d (tumor-associated macrophages)	-	‣ Activated by IL-6 or both TLR ligands and A2 adenosine receptor agonists [205,206] ‣ Secretes high levels of VEGF, IL-10 and TGF-β and downregulates TNF-α, IL-12 and IL-1β [207,208]

TNF-α: tumor necrosis factor-alpha; PMN: polymorphonuclear leukocytes; IL: interleukin; TGF-β: transforming growth factor-beta; NO: nitric oxide; ROS: reactive oxygen species; MMPs: matrix metalloproteinases; ECM: extracellular matrix; CCL: chemokine (C-C motif) ligand; IGF-1: insulin growth factor-1; PDGF: platelet-derived growth factor; iNOS: inducible nitric oxide synthase; STAT3: signal transducer and activator of transcription 3; TLR: Toll-like receptors; VEGF: vascular endothelial growth factor.

**Table 3 ijms-24-00641-t003:** Contributions of mast cells to tissue repair.

Repair Event	Functions
Hemostasis	Release tryptase to deactivate clotting induced by thrombin-stimulated fibrinogen; while they can also express fibrin stabilizing factor (XIIIa) to strengthen cross-linking of fibrin fibrils [228] Produce plasminogen activator inhibitor 1, a potent inhibitor of fibrinolysis, upon their activation by complements (C5a) [238] Enhance vascular permeability via heparin-mediated increase in bradykinin [239]
Inflammation	Induce vasodilation and increase vascular permeability to enhance cellular influx [240,241] Secrete MCP-1 to recruit monocytes [16,242] as well as TNF-α, CXCL8, kinins and proteases to recruit neutrophils [241,243,244] Promote the release of human β-defensin-3, an antimicrobial peptide, from epidermal keratinocytes [245] MCs-secreted histamine increases PPRs (TLR-2 and Dectin-1) expression on keratinocytes and raises levels of GM-CSF and CXCL8 [245]
Re-epithelialization	MCs-derived tryptase facilitates interactions between keratinocytes and MCs via degrading ECM and triggering PAR-2 receptors on keratinocytes [246] MCs-released LTB4 enhance keratinocyte proliferation [247], while MCs-produced histamine inhibits it [248]
Granulation tissue formation	Promote fibroblast proliferation through secreting bFGF, IL-4 and VEGF [229,249] MCs-derived histamine and tryptase stimulate fibroblasts to release FGF-2 or FGF-7 [229]
Angiogenesis	MCs-derived tryptase degrades the basement membrane to allow for endothelial cells migration and proliferation [250] Secret a variety of pro-angiogenic mediators such as PDGF, VEGF, FGF-2, bFGF, ANG-1 and TGF-β [236,250]

TNF-α: tumor necrosis factor-alpha; CXCL8: C-X-C Motif Chemokine Ligand 8; PPRs: pattern recognition receptors; TLR: roll-like receptor; PAR2: protease-activated receptor 2; LTB4: leukotriene B4; bFGF: basic fibroblast growth factor; VEGF: vascular endothelial growth factor; PDGF: platelet-derived growth factor; ANG-1: angiopoietin-1.

## Data Availability

No new data were created or analyzed in this study. Data sharing is not applicable to this article.

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
