# Peer review of "Acute Inflammation in Tissue Healing"

_ijms, 2022, doi:10.3390/ijms24010641_

Round 1

Reviewer 1 Report

Review of manuscript ijms-2058553

Acute Inflammation in Tissue Healing

Authors

Amro M. Soliman , Daniel R. Barreda

The manuscript is devoted to description of  the inflammatory process role in wound repair. The paper initially summarize the healing steps. Then, the inflammatory process initiation and development is discussed. Moreover, the mechanisms of an inflammatory cells transmigration to the healing milieu were described. The role of cytokines in the regulation of cell recruitment was highlighted. Partricipation of different inflammatory cells (neutrophils, macrophages, dendritic and Langerhause cells, mast cells and T cells) in inflammation development and  wound repair was discussed. Finally the mechanism of the cessation of inflammation as well as effect of disregulation of this process on repair was described.    

The manuscript is the well written review. Furthermore, it could be interesting for the readers of International Journal of Molecular Sciences.

There are a few concerns with the study:

Major concerns

  1. Description of the repair phases in the introduction is very scarce and incomplete.
  2. Moreover, some important steps in description of the mechanisms of cell transmigration to the wound are missing (integrin activation as well as integrin dependent tight adhesion).
  3. The Authors stress the role of neutrophils and proteolytic enzymes release in retarding of healing, but They did not referred to elastase.  
  4. In table 1 influence of  some cytokines (except TGF beta1) on ECM metabolism was not mentioned.
  5. Role of TGF beta1 in regulation of fibroblasts to myofibroblasts  transformation within the wound was not mention in table 1 or within the text.
  6. Except  cytokines the role of other mediators of inflammatory process (histamine, prostaglandins leukotriens or others) on wound repair regulation (proliferation of fibroblasts, angiogenesis, ECM deposition) was not discussed.

Minor concerns

1. Typing mistakes should be corrected.

2. Some sentences should be rearranged.

Author Response

The manuscript is devoted to description of  the inflammatory process role in wound repair. The paper initially summarize the healing steps. Then, the inflammatory process initiation and development is discussed. Moreover, the mechanisms of an inflammatory cells transmigration to the healing milieu were described. The role of cytokines in the regulation of cell recruitment was highlighted. Partricipation of different inflammatory cells (neutrophils, macrophages, dendritic and Langerhause cells, mast cells and T cells) in inflammation development and  wound repair was discussed. Finally the mechanism of the cessation of inflammation as well as effect of disregulation of this process on repair was described.    

The manuscript is the well written review. Furthermore, it could be interesting for the readers of International Journal of Molecular Sciences.

We sincerely thank the reviewer for the constructive review and feedback that rectifies the manuscript to an acceptable level for publication.

Major concerns

1. Description of the repair phases in the introduction is very scarce and incomplete.

A detailed description of tissue repair phases was added (lines 41–70)

2. Moreover, some important steps in description of the mechanisms of cell transmigration to the wound are missing (integrin activation as well as integrin dependent tight adhesion).

Role of integrin in cellular transmigration was added (lines 185-189)

3. The Authors stress the role of neutrophils and proteolytic enzymes release in retarding of healing, but They did not referred to elastase.  

A discussion on the contribution of elastase to ECM degradation was added (lines 237-245)

4. In table 1 influence of  some cytokines (except TGF beta1) on ECM metabolism was not mentioned.

The influence of TNF-α, IL-1β, IL-6, IFN-γ and IL-10 on ECM metabolism was added to Table 1.

5.Role of TGF beta1 in regulation of fibroblasts to myofibroblasts  transformation within the wound was not mention in table 1 or within the text.

Information was added to Table 1.

6. Except  cytokines the role of other mediators of inflammatory process (histamine, prostaglandins leukotriens or others) on wound repair regulation (proliferation of fibroblasts, angiogenesis, ECM deposition) was not discussed.

Role of inflammatory mediators, including histamine, leukotriene, thromboxane and PG in the regulation of tissue repair was added (lines 135-146)

Minor concerns

  1. Typing mistakes should be corrected.
  2. Some sentences should be rearranged.

We have carefully reviewed the text and corrected grammatical and typing mistakes in addition to sentence structure where applicable.  

Reviewer 2 Report

The authors of this article present a detailed review about the contributions of key cellular and molecular components to the acute inflammatory process and the effective and timely transition toward activation of tissue repair mechanisms. The topic is interesting and scientifically sound, as underlined by the enormous and wide literature already published on this topic. For the same reason, unfortunately I cannot see the novelty or the additional contribution into the field when compared to previous book chapters or review articles like:

Cooke JP. Inflammation and Its Role in Regeneration and Repair. Circ Res. 2019 Apr 12;124(8):1166-1168. doi: 10.1161/CIRCRESAHA.118.314669. PMID: 30973815; PMCID: PMC6578588.

Sugimoto MA, Sousa LP, Pinho V, Perretti M, Teixeira MM. Resolution of Inflammation: What Controls Its Onset? Front Immunol. 2016 Apr 26;7:160. doi: 10.3389/fimmu.2016.00160. PMID: 27199985; PMCID: PMC4845539.

Martin P, Nunan R. Cellular and molecular mechanisms of repair in acute and chronic wound healing. Br J Dermatol. 2015 Aug;173(2):370-8. doi: 10.1111/bjd.13954. Epub 2015 Jul 14. PMID: 26175283; PMCID: PMC4671308.

I strongly encourage the authors add additional and novel content and points of discussions, like for instance the role of the modern tissue engineering and or regenerative medicine in tissue haemostasis/healing …. 

Author Response

The authors of this article present a detailed review about the contributions of key cellular and molecular components to the acute inflammatory process and the effective and timely transition toward activation of tissue repair mechanisms. The topic is interesting and scientifically sound, as underlined by the enormous and wide literature already published on this topic.

We sincerely thank the reviewer for the time and effort reviewing the manuscript, and we appreciate the valuable and constructive feedback provided.

For the same reason, unfortunately I cannot see the novelty or the additional contribution into the field when compared to previous book chapters or review articles like:

Cooke JP. Inflammation and Its Role in Regeneration and Repair. Circ Res. 2019 Apr 12;124(8):1166-1168. doi: 10.1161/CIRCRESAHA.118.314669. PMID: 30973815; PMCID: PMC6578588.

Sugimoto MA, Sousa LP, Pinho V, Perretti M, Teixeira MM. Resolution of Inflammation: What Controls Its Onset? Front Immunol. 2016 Apr 26;7:160. doi: 10.3389/fimmu.2016.00160. PMID: 27199985; PMCID: PMC4845539.

Martin P, Nunan R. Cellular and molecular mechanisms of repair in acute and chronic wound healing. Br J Dermatol. 2015 Aug;173(2):370-8. doi: 10.1111/bjd.13954. Epub 2015 Jul 14. PMID: 26175283; PMCID: PMC4671308.

I strongly encourage the authors add additional and novel content and points of discussions, like for instance the role of the modern tissue engineering and or regenerative medicine in tissue haemostasis/healing …. 

We agree with reviewer on the great value provided by the papers pointed at, and they have contributed significantly to the topic of our manuscript. However, we respectfully provide an update on the most recently characterized significant contributions of various components of acute inflammation to tissue repair within the last 5 to 6 years; after some of the referred papers were published (in 2015 or 2016). Also, we humbly present a comprehensive review of the literature, which offers a one-stop shop for information on this topic with a summary of recent developments in this rapidly evolving research area. We appreciate reviewer’s suggestion and added a section on current tissue engineering strategies managing chronic injuries with a focus on modern potential applications targeting the inflammatory milieu and immune system behaviour, such as M2 polarization, and how they can contribute significantly to healing chronic injuries (lines 415 to 447).  

Round 2

Reviewer 1 Report

Thank you very much for the correction of the manuscript.

Only one problem remains to be explained. General theory of the cells transmigration suggest that selectins are involved in rolling when integrins after activation participate in tight adhesion. The Authors in line 188 suggest that integrins participate in thetering and rolling. If The Authors found the data for this fact they should describe these data in details.

Author Response

We thank the reviewer for his time and consideration. The required information has been added to the text (lines 196 - 206).  

Reviewer 2 Report

I acknowledge the authors for addressing my comments.

Author Response

We sincerely thank the reviewer for his time and effort reviewing this manuscript.